# Cytoreductive Nephrectomy and Metastatic Renal Cell Carcinoma: State of the Art and Future Perspectives

**DOI:** 10.3390/medicina59040767

**Published:** 2023-04-15

**Authors:** Luigi Napolitano, Celeste Manfredi, Luigi Cirillo, Giovanni Maria Fusco, Francesco Passaro, Marco Abate, Roberto La Rocca, Francesco Mastrangelo, Lorenzo Spirito, Savio Domenico Pandolfo, Felice Crocetto, Davide Arcaniolo, Biagio Barone

**Affiliations:** 1Department of Neurosciences, Reproductive Sciences and Odontostomatology, University of Naples “Federico II”, 80131 Naples, Italy; 2Unit of Urology, Department of Woman, Child and General and Specialized Surgery, University of Campania “Luigi Vanvitelli”, 80131 Naples, Italy

**Keywords:** cytoreductive nephrectomy, renal cell carcinoma, metastatic renal cell carcinoma, target therapy

## Abstract

In the past decades, several treatments have been proposed for the management of metastatic renal cell carcinoma (mRCC). Among these, cytoreductive nephrectomy (CN) represents a controversial and open issue in the era of targeted therapy and novel immunotherapy with immune checkpoint inhibitors. Two important studies, CARMENA and SURTIME, analyzed therapy with sunitinib with or without CN, and immediate CN followed by sunitinib versus a deferred CN after three cycles of sunitinib, respectively. CARMENA showed the non-inferiority of sunitinib alone versus sunitinib plus CN, whereas SURTIME showed no difference in progression-free survival (PFS), but a better median OS among patients with deferred CN. Therefore, more prospective clinical trials and appropriate patient identification are necessary to support CN in this new scenario. This review provides a snapshot of the current evidence for CN in mRCC, discusses the management strategies, and offers perspectives on the direction of future research.

## 1. Introduction

Renal cell carcinoma (RCC) is the sixth most common cancer in males and the ninth most common cancer in females, accounting for 78,000 newly diagnosed cases in 2018 in the United States, representing the third leading cause of mortality amongst genitourinary malignancies [1]. Nowadays, radical nephrectomy is the gold standard for large and locally advanced RCC [2], while nephron sparing surgery is the preferred for smaller localized tumors with percutaneous tumor ablation recognized as a safe option [3,4,5,6]; metastatic RCC (mRCC) management is more demanding, with conflicting results reported. Indeed, 5-year survival rates for mRCC range from 0 to 20% [7,8,9]. Although different retrospective studies have reported strong evidence in favor of cytoreductive nephrectomy (CN) [10,11], a paradigm shift in the management of patients with mRCC has been recorded in recent years. Indeed, new randomized trials do not seem to show an overall survival (OS) benefit for patients undergoing surgery plus systemic therapy versus systemic therapy alone.

Cytoreductive nephrectomy (CN) or metastasectomy consists of the removal of the primary tumor mass in the presence of metastases and is supported by several pieces of evidence based on studies conducted in the pre-targeted and targeted therapy eras [12]. The most important benefits of debulking the primary tumor are to assist and stimulate the immune system in controlling residual disease and removing the source of potential new metastases [13].

Biologic, i.e., targeted, therapy, which comprises the use of immunotherapy-based frontline systemic options, from interferon alpha and interleukin-2 to the use of PD-1 and VEGFR inhibitors [14], represents the standard treatment for patients with mRCC. Although the hereditary forms of RCC are estimated to represent only 3–5% of all kidney cancers, in recent years the role of the Von Hippel–Lindau (VHL) gene has been strongly related to renal carcinogenesis. Moreover, it has been shown that the incidence of multifocal disease is considerably higher in the hereditary renal cancer population, such as VHL, compared with the sporadic population [15]. The loss of VHL functions is related to the overexpression of hypoxia-inducible factor (HIF) and pro-angiogenic growth factors, as well as vascular endothelial growth factor (VEGF), epidermal growth factor (EGF) and pigment epithelial-derived factor (PEDF). All factors previously reported are involved in increased cell proliferation, survival and angiogenesis [13,16]. This represents the premises and the bases that led to the introduction of targeted therapies, which are emerging as a novel strategy in the management of mRCC.

This review aims to provide a snapshot of the current evidence on CN and a perspective on the direction of future research.

## 2. Materials and Methods

Bibliographic search was performed in the MEDLINE (US National Library of Medicine, Bethesda, MD, USA), Scopus (Elsevier, Amsterdam, The Netherlands) and Web of Science (Thomson Reuters, Toronto, ON, Canada) databases in November 2022. Different combinations of the following keywords were used according to a free-text protocol: “cytoreductive nephrectomy”, “renal cell carcinoma”, “metastasis”, “debulking”. Only articles in English were included. Conference abstracts, case reports, case series and commentaries were excluded. Results were narratively reported and no quantitative synthesis of data was performed due to the high heterogeneity of the studies.

## 3. Cytoreductive Nephrectomy and Metastatic Renal Cell Carcinoma

The role of cytoreductive nephrectomy (CN) in patients with mRCC is still unclear, as the benefit depends on the patient’s clinical and symptom presentation [17].

Whatever the uncertain benefits of the surgical procedure, the risk of the surgical procedure must be taken into account. Table 1 shows the benefits of CN.

In a retrospective study by the American College of Surgeons National Surgical Quality Improvement Program (ACS NSQIP), registry data showed that CN has more postoperative complications compared to nephrectomy in localized RCC (7.8 versus 3.2%, respectively) [19]. Among others, the most reported complications were pulmonary, thromboembolic and bleeding events. Consistently, mortality rates were higher in mRCC patients undergoing CN than those with non-disseminated disease (3.2 versus 0.5%). However, patients with disseminated cancer were older and reported greater comorbidities than those with localized RCC. These baseline characteristics have to be taken into when reading results. Likewise, a study from the Surveillance, Epidemiology and End Results Program (SEER) registry database reported similar data when considering 30-day mortality. In the entire cohort of over 24,500 patients, 219 deaths occurred during the initial 30 days after surgery (0.9% TDM rate). Among those, the 30-day mortality rate for mRCC patients was 4.2% vs. 0.3–1.3% for those with localized disease. Indeed, in a logistic regression model, age (*p* < 0.001) and stage (*p* < 0.001) achieved independent predictor status [20].

In mRCC patients, CN has several advantages including a decline in disease burden and growth of new metastasis. CN may have the potential advantages of improving the patient’s performance status score and relieving and controlling symptoms such as pain, hematuria, constitutional symptoms, and paraneoplastic manifestations. In theory, CN would also reduce the number of potentially immunosuppressive cancer cells and could prevent complications during systemic treatment. CN removed several factors including pro-angiogenic factors, and immunological [21]. A recent observational study evaluated 317 patients with mRCC treated with CN. The study divides patients into three groups according to symptoms: general symptoms (27%), local symptoms (37%) and specific metastases (23%). After CN treatment, the proportion of general signs and symptoms resolved had increased to 43%, the proportion of general symptoms improved had risen to 71%. Regarding local symptoms resolved, the percentage had increased to 91%, while the percentage of local symptoms improved had risen to 95%. In this study, however, it is also necessary to analyze the risk of complications, which was around 37%, as well as the risk of major complications, which was around 10% [22]. Table 2 shows the current clinical trials involving cytoreductive nephrectomy.

The main guidelines on renal cancer suggest that initial CN should be offered to patients with mRCC with good-risk renal neoplasia where single or oligometastatic disease sites can be fully treated with focal therapy (metastatectomy, radiotherapy, ablation) or observed until disease site therapy is needed. In fact, according to the current European guidelines, CN it is not recommended in MSKCC prognostic model poor-risk patients [23].

## 4. Cytoreductive Nephrectomy and Targeted Therapy

Nowadays, targeted therapies have widely superseded immunotherapy as the first option in mRCC management. The most used included VEGF monoclonal antibody, tyrosine kinase inhibitors (TKIs) (pazopanib, sunitinib, bevacizumab) and the mammalian target of rapamycin (mTOR) inhibitor (temsirolimus) [24]. All these agents had better outcomes in treatment compared to the standard immunotherapy, such as interleukin-2 (IL-2) and interferon-α2b (INF-α2b). Additionally, targeted therapies were better tolerated than immunotherapy in terms of toxicity profile and systemic symptoms (loss of appetite, malaise, fever and diarrhea) [25,26]. CN has traditionally been considered a pivotal process in the overall treatment strategy and has been included in nearly all clinical trials on metastatic renal cancer. Even if the survival benefits of CN are known, the exact mechanisms are still unclear and, despite superior outcomes, the response rate remains limited [13]. Petrelli et al., in a meta-analysis involving twelve studies with a total of 39,953 patients, evaluated the prognostic role of CN in advanced mRCC disease treated with molecular agents, reporting a lower mortality risk (by more than 50%) in patients treated with CN and targeted therapy compared to targeted therapy alone [27]. Conti et al. similarly analyzed the survival outcomes of CN, before and after the introduction of targeted therapy, in mRCC patients, reporting an increase in CN procedures from 29% to 39% with median survival increased from 13 to 19 months in patients receiving CN during the targeted therapy era. A limited increase (4 vs. 3 months) was instead found in patients not receiving CN before and after the targeted therapy era [28]. A phase 3 RCT aiming to determine the benefit of CN followed by targeted therapy in mRCC patients compared to targeted therapy alone (the CARMENA trial) showed, in a total of 450 patients enrolled from 2009 to 2017 (226 patients who underwent CN followed by sunitinib versus 224 patients treated with sunitinib alone), the non-inferiority of sunitinib alone versus nephrectomy plus sunitinib, additionally reporting a longer median OS for CN plus sunitinib (19.8 months vs. 15.6 months) [29]. However, the CARMENA trial revealed important limitations: patients with intermediate or poor-risk disease, who were included in the trial, do not appear to benefit from cytoreductive nephrectomy; patients eligible for upfront surveillance or delayed systemic therapy strategy post-cytoreductive nephrectomy were not included; a significant crossover between the two arms of the study was found (17% of patients in the sunitinib-alone arm underwent CN while 7% of patients in the CN arm did not undergo cytoreductive surgery); the trial experienced slow accrual, enrollment of higher metastatic burden patients and premature closure. Overall, these limitations, added to the raised concerns regarding potential recruitment bias, limited the generalizability of results as the cohort of patients involved in the study was not representative of cytoreductive nephrectomy candidates in a real-world setting [30,31]. Another RCT (SURTIME), conducted by Bex et al., aimed to compare the outcomes between sunitinib before CN versus immediate CN followed by sunitinib. The study closed after 5.7 years, enrolling a total of 99 patients (80 men and 19 women). The 28-week progression-free rate (PFR) was 42% in the immediate CN arm (n = 50) and 43% in the deferred CN arm (n = 49) (*p* = 0.61). Patients with deferred surgery reported improved survival compared to patients who underwent upfront nephrectomy (hazard ratio of 0.57, 95% CI 0.34–0.95; *p* = 0.03). With the deferred approach, more patients received sunitinib, yielding improved overall survival (OS) and permitting early detection of those who showed inherent resistance to systemic therapy before CN. Deferred CN would therefore be a reasonable option in those with intermediate-risk disease (in which a delayed therapy could be detrimental) or in patients with aggressive disease who are non-responders to targeted therapy [32]. To date, no study has prospectively validated the role of CN with targeted therapies and international guidelines (EAU and NCCN) recommend the use of CN only from extrapolated favorable outcomes [23].

Lastly, over the last decade, significant progress has been made in identifying the molecular characterization of mRCC, leading to an increased understanding of targeted therapy [33]. However, the individual efficacy of currently available drugs remains unpredictable [34]. Indeed, combination therapy using TKIs and immune checkpoint inhibition (ICI) proved to be superior to TKI monotherapy with sunitinib in three pivotal phase 3 studies [35]. Prolonged follow-up data including subsequent therapy lines reported significant overall survival benefit for patients with upfront combination regimens. In contrast, primary resistance against TKI occurring in approximately 30% of patients is known to result in worse prognosis due to limited response to successive therapy with VEGF-targeted TKIs or mTOR inhibitors [36,37]. A future prospective study could determine if primary resistance to first-line TKI monotherapy predicts the response to ICI in subsequent therapy lines and influences survival outcomes.

## 5. Cytoreductive Nephrectomy and Cytokines

In the era of cytokine therapy, the role of CN practiced before systemic therapy was widely studied, as, in relation to RCC, it was found that large tumors blocked elements of the immune system, a key player in the fight against cancer [38,39]. In favor of this thesis, a well-conceived study demonstrated the spontaneous regression of metastatic lesions after nephrectomy [39]. Therefore, it was found that practicing CN could unlock the immune system and increase the beneficial effect of cytokine therapy.

The Southwest Oncology Group (SWOG) and the European Organization for Research and Treatment of Cancer (EORTC) published two major studies in 2001 on patients with mRCC who were treated with and without cytokine therapy [40,41]. The primary endpoint of the study was to evaluate the survival and then the radiographic response of the metastatic disease. Both studies had equal recruitment criteria including operable primary tumors and good performance status (PS). SWOG 8949 selected 98 patients who were treated with CN plus IFN-α and 121 patients treated with IFN-α alone. Both groups gave similar responses, except for a 3-month longer survival in patients treated with CN plus IFN-α compared to patients treated with IFN-α alone (11.1 vs. 8.1 months, *p* = 0.05). EORTC 30947 instead selected 85 eligible patients, of which 42 were treated with CN plus IFN-α and 43 were treated only with IFN-α. The radiographic regression response of metastatic disease in both groups was similar, while patients undergoing the combined treatment enjoyed significantly longer survival (17 vs. 7 months, *p* = 0.03).

These two studies were combined in 2004 by Flanigal et al. The combined analysis of the two studies showed an overall survival advantage of 5.8 months (13.6 vs. 7.8 months, *p* = 0.002) and a 31% reduction in the risk of death for patients undergoing cytoreductive nephrectomy regardless of the prognostic factors considered (performance status, disease sites and extent of illness). Peri-operative mortality associated with cytoreductive nephrectomy was 1.5% while severe peri-operative complications were equal to 5% [42]. Nevertheless, several shortcomings and limitations of previously reported trials led to two major controversies regarding the role of CN in mRCC management. Firstly, PS 1 patients assigned to treatment groups were unequally divided, with 58.9% in the immunotherapy arm and 46.6% in the CN arm. Additionally, PS 1 is associated with a worse prognosis compared to PS 0, with a median survival of 6.7 months versus 10.1 months, respectively [43]. Secondly, both trials were significantly underpowered, thus limiting the validity of results and conclusions. Lastly, the eligibility criteria forced the recruitment of patients with a PS of 0 or 1 in addition to a resectable primary tumor [44]. As a result, data obtained were limited for other categories of patients, i.e., unresectable primary tumor, disseminated disease or multiple metastases. Despite these limitations and shortcomings, the role of CN in the treatment of mRCC has been supported by several, albeit retrospective, studies [13]. More recently, a Cochrane-based analysis by Coppin et al. involving 6880 patients with advanced RCC undergoing immunotherapy in a total of 58 studies, confirmed, in selected surgical patients, the role of CN prior to immunotherapy as the best survival strategy [45].

## 6. Immediate vs. Deferred Cytoreductive Nephrectomy

Nowadays, the immediate vs. deferred cytoreductive nephrectomy question remains unresolved. Recently, two important trials, CARMENA and SURTIME, have introduced novel elements regarding the treatment of mRCC [27,30]. The CARMENA trial aimed to investigate the role of CN in the targeted therapy era, while SURTIME aimed to determine whether deferred CN in combination with sunitinib could be used to identify patients that were non-responders to targeted therapy. CN has been historically performed in oligometastatic patients in addition to complete metastasectomy [27]. Different hypotheses have been reported to justify the beneficial effect of CN: first of all, CN decreases the production of growth factors and cytokines by the primary tumor and this postponed metastatic progression; secondly, the nephrectomy-activated azotemia. CARMENA aimed to study the role of immediate CN followed by sunitinib versus sunitinib alone. This trial showed the non-inferiority of sunitinib in comparison with CN followed by sunitinib. After 50.9 months of follow-up, the ITT analysis showed a median OS with CN of 13.9 months versus 18.4 months with sunitinib alone (HR 0.89; 95% CI: 0.71–1.10) while in intermediate-risk patients, median OS was 19.0 months with CN plus sunitinib versus 23.4 months of sunitinib alone (HR 0.92; 95% CI: 0.60–1.24). Lastly, poor-risk patients reported an OS of 10.2 months versus 13.3 months, respectively (HR 0.86; 95% CI: 0.62–1.17). The median PFS in the ITT group was instead 7.2 months (CN) versus 8.3 months (sunitinib) (HR 0.82; 95% CI: 0.67–1.00). SURTIME is a randomized clinical trial that investigated if the use of sunitinib therapy before CN improved outcome compared with immediate CN followed by sunitinib [30]. Progression-free survival (PFS) was the primary endpoint, while OS, adverse event rates and postoperative progression were the secondary endpoints. No difference was found in the PFS, while the OS hazard ratio (HR) of deferred versus immediate CN was 0.57 [95% confidence interval (CI): 0.34–0.95, *p* = 0.032], yielding a median OS of 32.4 (95% CI: 14.5–65.3) and 15.0 months (95% CI: 9.3–29.5), respectively. These two trials suggested that immediate CN does not show benefits in patients with primary clear cell mRCC requiring sunitinib. According to SURTIME, a deferred CN should be proposed in patients starting with sunitinib with a non-progressive disease. According to these studies it is mandatory to select the patients based on therapy response, because CN results in higher morbidity and mortality compared to nephrectomy [46].

## 7. Selection of Patients

Despite recent advances, the optimal candidates undergoing CN remain open to discussion. Zhang et al., in a study involving a total of 5544 patients from the Surveillance, Epidemiology and End Results (SEER) database, of which 2352 (42.4%) underwent CN, showed that surgical outcomes were affected by pathological grade as well as the number of distant metastases [47]. In particular, pathological Grade 4 was associated with a bad prognosis, and the presence of a major organ metastasis (bone, brain, liver and lung) improved survival time. Zahng et al. also showed that, in 6043 patients with mRCC, CN was not suggested as the first-line strategy in T4 patients; in particular, T4 stage, N1 and age  ≥  76 yr were important risk factors influencing CSS [48]. Faba et al. reported that CN showed several benefits in young male patients with oligometastatic disease and a good performance status, while patients whose life expectancy was short, who had poor risk-IMDC (International Metastatic RCC Database Consortium), or with liver and/or bone metastases, and positive lymph nodes, in addition to sarcomatoid components, showed no benefits from CN [49]. In particular, the presence of lymphadenopathy seems to be a significant negative prognostic factor in patient stratification. Bing Ji et al. reported their experience in 1229 patients with metastatic sarcomatoid RCC, showing an improved OS with CN in T1 and T2 patients. Due to this, further prospective and randomized trials are necessary to assess a nomogram to better identify and stratify patients suitable for CN [50].

## 8. Future Perspectives

New CN studies are complex and depend on the rapid evolution of therapeutic algorithms. In addition, as shown in the CARMENA and SURTIME trials, there is a difficulty in patient enrollment: both trials took 8 years to collect patients but did not achieve full enrollment. With new therapeutic options replacing sunitinib, which now represent the gold standard in the treatment of mRCC, the continuation of these trials would have resulted in ethical dilemmas and both trials were discontinued. For any future trials, it must also be considered that the number of eligible patients is much lower than 20 years ago due to the changed epidemiology as well as the high probability that the first choice of therapy will change during the trial. Additionally, considering the controversial results of previous trials, there is certainly no need to conduct large studies in this very selective population, but new studies could be scaled down by trying to highlight the benefits of delayed versus no CN. Alternatively, biomarkers could be sought in order to permit the calculation of smaller sample sizes based on higher HRs with the aim of delivering faster read-outs of study results.

## 9. Conclusions

The past 20 years have seen a transition in the management of mRCC from a primary focus on CN to an increased emphasis on multimodal therapy. This evolution has been driven, in a large part, by the rapid development of systemic therapeutic agents. Several observational and randomized studies have attempted to elucidate the role of CN in the era of targeted therapies, but controversy remains. There seems to be a need for careful patient selection and an increasingly central role in multidisciplinary discussion. The use of CN in mRCC could enable a more developed model of tailored oncology, permitting the early identification of patients who could best benefit from therapy.

## Figures and Tables

**Table 1 medicina-59-00767-t001:** Benefits of CN.

Regression of metastases [10]
Reduced release of cytokines and growth factors promoting metastatic spread [12,13]
Prevention of tumor growth and new metastases [11,12,13]
Relief from symptoms (i.e., hematuria, pain)
Reduced paraneoplastic syndromes [18]
Improvement in QoL

QoL = quality of life.

**Table 2 medicina-59-00767-t002:** Current ongoing trials involving cytoreductive nephrectomy.

NCT Number	Other Name	Interventions	Phase	Outcome Measures
NCT05327686	SAMURAI	Avelumab, Axitinib, Cabozantinib, Ipilimumab, Lenvatinib, Nivoluma, Pembrolizumab, Stereotactic Ablative Radiotherapy	2	Progression-free survival
NCT04510597	PROBE	Cytoreductive Nephrectomy	3	Overall survival
NCT04370509	-	Axitinib, Cytoreductive Nephrectomy, Metastasectomy, Pembrolizumab		Proportion of participants with ≥2-fold increase in the number of tumor-infiltrating immune cells
NCT04322955	Cyto-KIK	Cabozantinib, Nivolumab, Cytoreductive Nephrectomy	2	Percentage of participants with a complete response
NCT03977571	NORDIC-SUN	Cytoreductive nephrectomy	Not available	Overall survival
NCT03324373	-	Lenvatinib, Everolimus, Cytoreductive Nephrectomy	1	Surgical complications
NCT00715442	-	Sunitinib, Cytoreductive Nephrectomy	2	Time to progression

## Data Availability

The data presented in this study are openly available.

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
