# Peer review of "Cytoreductive Nephrectomy and Metastatic Renal Cell Carcinoma: State of the Art and Future Perspectives"

_medicina, 2023, doi:10.3390/medicina59040767_

Round 1

Reviewer 1 Report

It was a pleasure reviewing the manuscript "Cytoreductive nephrectomy and metastatic renal cell carcinoma: state of art and future perspectives".

Table 1 -- It is unclear what is meant by "Implementation of patient PS to allow more compliant adjuvant therapies"-- adjuvant will be in the setting of no evidence of disease. 

Table 1-- please provide reference of "Reduced paraneoplastic syndromes" is observed with cytoreductive nephrectomy

In section 3.  what is PN?

Section 3 "In fact, according to the current European guidelines, it is recommended with a strong non-perform rating CN in MSKCC prognostic model poor-risk patients." I am not sure what is strong non-perform. Also please add citation.

Section 6. I do not think authors are correctly interpreting SURTIME study.

It is quite confusing to understand what the authors feel critically and what readers should do.

Reviewer 2 Report

This is a nice manuscript recapitulating a hot topic in the field of urological oncology. Since decades there is a discussion ongoing whether cytoreductive surgery in metastatic renal cell carcinoma is a way that have to be run. This manuscript gives an overview to all interests who are not so familiar with this argument. 

But some comments have to be done

Introduction

The authors cite 78000 new cases in 2018. But nobody knows to which region of the world it refers to.

Paragraph 3, line 2

(metti referenza) has to be replaced by a true reference

Paragraph 4

It is very verbose and difficult to read. This paragraph needs a better arranged structure. 

Round 2

Reviewer 1 Report

It was a pleasure reviewing the manuscript. The authors have replied to my queries upto my satisfaction. I do not have any further questions.